# Antimicrobial Activity of *Sempervivum tectorum* L. Extract on Pathogenic Bacteria Isolated from Otitis Externa of Dogs

**DOI:** 10.3390/vetsci10040265

**Published:** 2023-03-29

**Authors:** Diana Maria Dégi, Kálmán Imre, Viorel Herman, János Dégi, Romeo Teodor Cristina, Adela Marcu, Florica Morariu, Florin Muselin

**Affiliations:** 1Department of Toxicology and Toxicoses, Plant Biology and Medicinal Plants, University of Life Science “King Mihai I” from Timișoara, 300645 Timisoara, Romania; 2Department of Public Health, University of Life Science “King Michael I” from Timișoara, 300645 Timisoara, Romania; 3Department of Infectious Diseases and Preventive Medicine, University of Life Science “King Mihai I” from Timișoara, 300645 Timisoara, Romania; 4Department of Pharmacology and Veterinary Pharmacy, University of Life Science “King Mihai I” from Timișoara, 300645 Timisoara, Romania; 5Department of Animal Production Engineering, Faculty of Bioengineering of Animal Recourses, University of Life Science “King Mihai I” from Timișoara, 300645 Timisoara, Romania

**Keywords:** *Sempervivum tectorum* L., bacteria, otitis externa, dog, phytopharmacology

## Abstract

**Simple Summary:**

Two distinct but strongly related aims directed our work. First, we aimed to evaluate the phytotherapeutic potential of *Sempervivum tectorum* L., a traditionally well-known but chemically less characterized medicinal plant, through quantitative analysis of its biologically active substances. Modern methods of phytoanalytics (microwave extraction, lyophilization, and UV spectrophotometry) were applied to investigate the polyphenol and proanthocyanidin contents of *Sempervivum tectorum* L., which have significant biological and pharmacological activities, including antimicrobial activity. Our second aim was to determine the antimicrobial activity of *Sempervivum tectorum* L. extract against pathogenic bacteria isolated from ear swabs taken from dogs with with chronic and recurrent otitis externa and standard ATCC strains (respectively, *Staphylococcus aureus* ATCC 25923 and *Pseudomonas aeruginosa* ATCC 27853). *Sempervivum tectorum* L. represents a particular model plant selected to represent the wide structural and therapeutical variety of phenolics, from simple phenolic acids to macrocyclic polyphenols.

**Abstract:**

The present study investigated the antimicrobial activity, total phenolic content, and proanthocyanidin concentration of ethanolic extracts from fresh leaves of *Sempervivum tectorum* L. The extracts were phytochemically analyzed and evaluated for antimicrobial activity. The broth microdilution method was used to assess antimicrobial activity against pathogenic bacteria isolated from ear swabs taken from dogs with otitis externa. Many compounds were present in the ethanolic aqueous extracts, which exhibited a broad spectrum of antimicrobial activity. They showed strong antibacterial activity against standard clinical Gram-positive strains such as *S. aureus* and Gram-negative strains such as *P. aeruginosa*. In our study, the obtained quantity of total phenolic compounds in the ethanol:water extract of leaves was 126.17 mg GAE/g. The proanthocyanidin concentration in the tested *Sempervivum tectorum* L. extracts was 15.39 mg PAC/g material. The high contents of total phenolics and proanthocyanidin indicated that these compounds contribute to antimicrobial activity. The antimicrobial activity of the tested *S. tectorum* L. extracts ranged from 1.47 to 63.75 µg/mL, starting with 1.47 µg/mL and 1.75 µg/mL against *S. aureus* ATCC 25923 and *P. aeruginosa* ATCC 27853 strains, respectively. Likewise, *S. tectorum* L. ethanol extract demonstrated a bacteriostatic effect against *S. aureus* clinical isolate with a median MIC of 23.25 µg/mL and MBC of 37.23 µg/mL; and bactericidal against *S. aureus* ATCC 25923 with the median MIC of 20.33 µg/mL and MBC of 37.29 µg/mL. In the Gram-negative *P. aeruginosa* clinical and standard strains, the expressed MIC and MBC values were 24.234 and 20.53 µg/mL for MIC, and 37.30 and 37.02 µg/mL for MBC, respectively.

## 1. Introduction

Plants have long been essential sources of therapeutic agents for both humans and animals [1,2,3]. The resurgence of interest in the pharmacological activities of plants, in recent decades has fueled the development of standardized methods of preparation and extraction [4].

Traditional veterinary care using plants is called ethnoveterinary medicine, and it is practiced widely across Europe, including in Romania [3,4,5,6,7,8,9].

Herbal remedies have traditionally been administered in the context of a holistic approach to health maintenance and well-being that integrates enhancement of various body processes with symptom relief [10]. *Sempervivum tectorum* L. is an evergreen plant from the *Crassulaceae* family [11,12,13]. *S. tectorum* L. is widely recognized in traditional veterinary medicine, and its juice is still employed as a cooling agent and astringent to treat wounds, burns, insect bites, and the treatment of ear infections in dogs [13,14,15,16]. 

*S. tectorum* L., known as the common houseleek, is a thorny perennial herb with a dense basal rosette of fleshy obovate to oblong, sharp-tipped, bluish-green leaves [11,14,15]. The flowers are arranged in terminal cymes and are a dull red color. Houseleek is native to central and southern Europe and grows on weathered rocks, roofs, and scree. Traditionally, the leaves are used fresh rather than dried.Houseleek contains tannins, bitter compounds, sugars, flavonoids, and mucilage [17]. The plant has been linked to antinociceptive, liver-protective, and membrane-stabilizing effects, which have been partly attributed to the antioxidant activity of its phenolic compounds [18,19]. It has numerous external applications: the pressed juice of the leaves can be applied to herpetic skin eruptions, suppurating wounds, inflamed insect bites, and external ear inflammation in dogs [15,17]. Romania’s flora consists of approximately 4000 species, 800 of which have therapeutic characteristics validated by scientific investigations in over fifty percent of cases [7,8,20]. Extraction by conventional method relies on heating the solvent using the open microwave-assisted extraction (MAE) method, which involves contact with the sample matrix and disrupts cellulosic cell walls in plants, allowing easy acquisition of active components [21,22,23,24]. Otitis externa (OE) in dogs, a common veterinary complaint, can be caused by primary, secondary, predisposing, or perpetuating factors [25,26]. Chronic or recurrent otitis externa in dogs is more difficult to treat, although acute and uncomplicated otitis externa in dogs can frequently be successfully managed [27]. Antimicrobial, antimycotic, and anti-inflammatory medications are used as pharmacological support, but treatment failure and antimicrobial resistance are coercing the development of alternative approaches based on phytotherapeutic agents [28]. The most common bacteria isolated in canine OE cultures are *Staphylococcus pseudintermedius*, *P. aeruginosa*, *S. aureus*, *Enterococcus* spp., *Corynebacterium* spp., *Streptococcus* spp., and *Escherichia coli* [27,28,29,30,31]. Colonization by opportunistic bacteria, yeast (such as *Malassezia*), and antimicrobial resistance could complicate the pathology and can pose a therapeutic challenge for veterinarians. Plant bioactive compounds are commonly used in veterinary medicine, and some have been found to have antibacterial effect against pathogenic bacteria isolated from clinical cases of canine otitis externa [32,33]. Given the increase in multi-antimicrobial-resistant bacteria, which is likely owing to selective pressure resulting from excessive/inappropriate antimicrobial use, this is a fascinating concept [34]. Unlike antibiotics, bioactive substances work at the point of application and are less prone to produce resistance when used often [32]. An increasing antimicrobial resistance in the *Staphylococcus* spp. and *P. aeruginosa*, isolated from dogs’ ear clinical samples, has been reported [27,35]. Typically, both bacterial species are known as common dog colonizers but are also considered opportunistic pathogens that can cause various diseases [36]. Methicillin-resistant *S. aureus* (MRSA) and multi-drug-resistantant *P. aeruginosa* in healthcare settings have become a global concern for the health of dogs and humans [27,37]. They are typically linked with a multidrug-resistant phenotype, limiting veterinarians’ therapeutic options [38]. 

This study aimed to evaluated the antibacterial activity of extracts of *S. tectorum* L. samples against isolated bacteria and in standard strains (*S. aureus* and *P. aeruginosa*) via chemical analysis using UV spectrophotometry. The antibacterial activity, total phenolic content, and concentration of flavonoids of ethanolic extracts of fresh *Sempervivum tectorum* L. leaves were studied.

## 2. Materials and Methods

### 2.1. Study Area 

*Sempervivum tectorum* L. plant materials were collected from spontaneous flora in the Socolari area (GPS Coordinates: 44°56’39,57” N, 21°43’45,00” E; Appendix A), in the mountain region of western Romania, during spring and summer 2019. 

*S. tectorum* L. samples and herbarium specimens were deposited at the Department of Vegetal Biology and Medicinal Plants, where sample authentication was also accomplished (voucher no. 112). The plants were identified according to the determination key given by Flora *Europaea* (*Sempervivum tectorum LINN.;* Sp. Plant. 464/1753), based on information provided by the Euro+Med PlantBase [39].

### 2.2. Chemicals and Plant Material

Ethanol, sodium hydrogen carbonate, triphenyl tetrazolium chloride (TTC), and Folin–Ciocalteu phenol reagent were acquired through from Sigma Aldrich (Merck KGaA, Darmstadt, Germany). Agar used for microbiological exams was obtained from Becton Dickinson (Becton Dickinson GmbH, Heidelberg, Germany). All other solvents and chemicals met analytical standards. The leaves were removed from the rosette and roots, washed, and filtered through grade 4 Whatman^®^ qualitative filter paper (Merck KGaA, Darmstadt, Germany) before lyophilization (Appendix A). The samples were lyophilized to obtain the crude mass of juice for further experiments. 

#### Lyophilization Method

The material was lyophilized using the Leybold Heraeus Lyovac GT2 (LH Leybold, Labexchange–Die Laborgerätebörse GmbH, Burladingen, Germany). The lyophilized *S. tectorum* L. leaves were then finely powdered in a blender as the pressure was slowly brought back up to room temperature and pressure. When a heat-dependent drying procedure could potentially denature thermosensitive components present in a material, lyophilization is the preferred method of drying [40,41].

### 2.3. Conventional Extraction Method

#### 2.3.1. Sample Preparation for Conventional Extraction Method

##### Microwave-Assisted Extraction (MAE) 

Each lyophilized sample (5 g) was placed in a round-bottom flask, followed by 50 mL of solvent (aqueous ethanol, 50:50 *v:v*). MAE was performed for 5 min at 50 W and 35 °C on a CEM Star 2 Plus Open Vessel Microwave Digestion System (CEM Corporation, Matthews, NC, USA), and the extract was centrifuged at 4000 rpm for 10 min and filtered through grade 4 Whatman^®^ filter paper (two more measures of solvent were used to extract the press residue (total solvent volume: 150 mL)). Before further analysis, at 40 °C, the mixed extract was evaporated until dry and then redissolved in 30% solvent [17,42,43]. The *S. tectorum* L. leaves were ground to powder in a blender and subjected to MAE. After a specified length of selected conventional extraction, the filtrates were filtered and collected. A rotary evaporator (Julabo-Pura 4 water bath) was used to remove the solvent from the filtrate (Julabo UK Ltd., Stamford, UK). The extracts were stored at −20 °C until further examination.

#### 2.3.2. Analytical Methods

##### Total Phenolic Content

Spectrometric quantification of polyphenols and proanthocyanins was carried out in accordance with the European Pharmacopeia directive.The total phenolic content (TPC) of extracts was determined by UV spectrophotometry (Varian Cary 50 UV-VIS, Agilent Technologies, Santa Clara, CA, USA) using the Folin–Ciocalteu (FC) method described by Anokwuru et al. [44]. Gallic acid (GA) equivalent (mg GA/g raw material) was used to express TPC. Total polyphenols were measured using a wavelength of 241 nm and compared to ethanol as a blank.To determine total phenolics using the FC technique [45], 800 mL of sodium carbonate (7.5 percent *w/v*) was added to 20 mL of each sample (range from 0.5 to 20 mg/mL), properly mixed, and allowed to stand for 2 min. Then, 1 mL of FC reagent was added while the mixture was vortexed. After leaving the samples in the dark for 30 min at room temperature, the absorbance was measured spectrophotometrically at 765 nm. 

##### Proanthocyanin Content

According to Mannino et al. [46], we used UV spectrophotometry to determine the proanthocyanin (PAC) contents in extracts (Varian Cary 50 UV-Vis; Agilent Technologies, Santa Clara, CA, USA) based on acid hydrolysis and color formation.

### 2.4. Antimicrobial Activity Testing

Extracts were evaluated against a panel of bacteria, which included Gram-positive *S. aureus* clinical isolate and ATCC 25923, and Gram-negative *P. aeruginosa* clinical isolate and ATCC 27853, which were collected and cultured from ear swabs of dogs with chronic and recurrent otitis externa (Appendix A). The clinical strains are multi-drug- and methicillin-resistant (Appendix A). Clinical Laboratory Standards Institute (CLSI) [47,48] standards were followed for culture, identification, and susceptibility testing. After investigating the morphological and biochemical characteristics using standard laboratory methods reported and recommended by *Bergey’s Manual of Systematic Bacteriology*, the purified bacterial cultures were identified and confirmed (Appendix A). Clinical breakpoints of gentamicin (clinical breakpoint range: 0.5–8 g/mL, respectively 0.5–16 g/mL) and enrofloxacin (clinical breakpoint range: 0.5–4 g/mL, respectively 0.25–16 g/mL) for *P. aeruginosa* and *S. aureus* were used, taking into account the body site and host [49,50].

The bacterial isolates used were clinical isolates from the microbiological collection of the Microbiology Laboratory of the Faculty of Veterinary Medicine, Department of Infectious Diseases and Preventive Medicine, King Michael I University of Life Sciences, Timișoara. Antibacterial activity was assessed utilizing the CLSI M07-A9 broth microdilution method [51]. The minimum inhibitory concentration (MIC) was determined using serial dilution in 96-well microdilution plates (capacity: 200 μL). Mueller–Hinton agar for bacteria (Becton Dickinson GmbH, Heidelberg, Germany) was used to culture the test species at 37 °C. The final bacterial inoculum density was 5 × 10^5^ CFU/mL [52].

To achieve a concentration of 1 mg/mL of lyophilized extract, stock solutions of *Sempervivum* extract were prepared in 70% ethanol (20 mg lyophilized extract dissolved in 250 mL ethanol to obtain an 80 g/mL concentration). The extracts were then serially diluted in concentrations ranging from 0.5 to 62 g/mL (0.5, 1.0, 2.0, 4.0, 8.0, 16.0, 32.0, and 64.0 g/mL). The ethanol concentration in each well was never higher than 5%. The inoculum was added to each well, and the plates were cultured for 24 h at 37 °C. Positive controls included enrofloxacin (ENR, 5 µg) and gentamycin (GN, 10 µg) (both from Bio-Rad, Marnes-la-Coquette, France), while the solvent acted as a negative control. One inoculated well was included to allow for control of the broth’s suitability for organism growth. To ensure sterility of the medium, one non-inoculated well free of antimicrobial agents was also included (sterility control). The growth of bacteria was determined by adding 20 μL of 0.5% triphenyl tetrazolium chloride (TTC) aqueous solution (growth control). The minimal inhibitory concentration (MIC) was established as the lowest concentration of extract that inhibited observable growth (a red-colored pellet on the bottom of the well after the addition of TTC). To measure the minimal bactericidal concentration (MBC), broth from wells without apparent growth was inoculated at 37 °C for 24 h with Mueller–Hinton agar (Becton Dickinson GmbH, Heidelberg, Germany).

### 2.5. Statistics 

The data represent the standard deviation of the mean of three replicates (SD). Using SPSS version 20.0, the results were subjected to multiway analysis of variance, and mean comparisons were performed using Tukey’s multiple range test (SPSS Inc., Chicago, IL, USA). Mean differences were considered significant at *p* < 0.05.

## 3. Results

Lyophilized *S. tectorum* L. from the mountain region was used. Freeze-drying was used since it is highly efficient and does not require high temperatures.

### 3.1. Total Phenolic Composition

Polyphenolic compounds are aromatic hydroxylated compounds that are present in a wide variety of plant species. Flavonoids, anthocyanidins, and resveratrol are the most well-known polyphenols. The total phenolic content was determined primarily based on the extracts’ antimicrobial activity. Polyphenols from *S. tectorum* L. can potentially contribute to research as components of treatments for well-known diseases such as otitis externa [16]. Table 1 shows the total phenolic content (TPC) of the *S. tectorum* L. extracts obtained using microwave-assisted extraction in this study.

The lyophilized *S. tectorum* L. samples had the highest TPC (126.17 mg GA/g material). Our results show that using the microwave-assisted extraction method and a 1:1 ethanol:water mixture was the best way to extract the highest TPC. This is ideal, because these solvents are preferable in plant compound extract applications. Microwave-assisted extraction with ethanol as a solvent yielded the highest total phenol content (126.17 mg GA/g material). The TPC can be influenced by the climatic conditions under which a plant is grown. This could be why the amounts of certain active substances in plants can vary so greatly. Another reason for the large differences reported in total phenolic contents could be the preparation process. Certain active compounds disintegrate when samples are dried at high temperatures, but most compounds retain their original activity when lyophilized because the process does not use high temperatures. We can confirm that the lyophilization process is the best option for sample preparation. Fresh biomass has the advantage of requiring minimum processing and resulting in minimal degradation of the target bioactive components. The issue is that it can include more than 70% water, diluting the extraction solvent. When preparing raw materials, water is also an issue because it impedes shredding and grinding. Even if the herb is in bulk, grinding it to the desired size prior to extraction is easier with dry biomass. Furthermore, the solvent/herb ratio is much easier to control, providing greater reproducibility in experimental work. As a result, using a microwave technique for plant extraction has benefits such as enhanced mass transfer, improved solvent penetration, less reliance on the solvent utilized, extraction at lower temperatures, faster extraction rates, and higher product yields [53,54]. 

### 3.2. Proanthocyanidin (PAC) Content

Proanthocyanidins (PACs) are oligomeric or polymeric polyphenols that are found in many plants. They mainly comprise catechin, epicatechin, and gallic acid ester oligomers. They have antibacterial properties, just like polyphenols [55]. However, their biological abilities are primarily determined by their structures [56].

The PAC content of the *S. tectorum* L. extracts is shown in Table 1. The highest PAC content (15.39 mg PAC/g material) was obtained in lyophilized *S. tectorum* L. material using microwave-assisted extraction and a 50:50 ethanol:water mixture.

The analysis revealed extremely high PAC content in lyophilized *S. tectorum* L. extracts; because the active components did not disintegrate, the lyophilization process was ideal for preparing the studied materials.

### 3.3. Antimicrobial Activity

The microdilution method was used to evaluate the minimum inhibitory concentration (MIC) and minimum bactericidal concentration (MBC) of the most effective *S. tectorum* L. extract. Table 2 and Table 3 show the concentration effect of this extract (Appendix A). The MIC and MBC of this *S. tectorum* L. extract indicated high activity. The ethanolic leaf extract demonstrated the highest activity, with determined inhibitory activity against all tested strains.

The inhibitory effects of the *S. tectorum* L. extract became observable at 1.47 µg/mL against *S. aureus* ATCC 25923 and 1.75 µg/mL against *P. aeruginosa* ATCC 27853. 

Antibacterial activity was determined according to the MBC/MIC ratio (Table 2). If the ratio was less than 4, the effect was considered bactericidal; if it was greater than 4, it was considered bacteriostatic. The *S. tectorum* L. ethanol extracts were bacteriostatic against the *S. aureus* clinical isolate strain, with a median MIC of 23.25 µg/mL and MBC of 37.23 µg/mL, and were bactericidal against *S. aureus* ATCC 25923, with a median MIC of 20.33 µg/mL and median MBC of 37.29 µg/mL. The median MIC and MBC of the Gram-negative *P. aeruginosa* clinical and standard strains were 24.234 and 20.53 µg/mL and 37.30 and 37.02 µg/mL, respectively.

The MBC was confirmed by the absence of bacterial growth in the inhibition zone corresponding to the tested strain’s lowest MIC. With an MBC of 15.75 µg/mL, *S. tectorum L*. extract demonstrated potential bactericidal activity against the tested pathogenic bacteria (*S. aureus* and *P. aeruginosa*) (Table 3). The MIC and MBC results of the most effective plant extract suggest that *S. tectorum* L. can be used to control and prevent the growth of bacteria implicated in the pathogenesis of otitis externa in dogs. The bacterial strains were chosen for their importance in the pathogenesis of canine otitis externa. *P. aeruginosa* is one of the most common pathogenic bacteria involved for otitis externa in dogs. Toxins and other metabolites produced by *S. aureus* cause skin lesions, a risk factor for otitis externa in dogs. *S. tectorum* L. extract inhibited the growth of all tested bacterial strains.

Because the Kruskal–Wallis *p*-value was less than the significance level (c2 = 12.222, *p* < 0.001), we concluded that there were significant differences in total phenolic and proanthocyanidin contents in extracts obtained using the microwave-assisted extraction method: S. *tectorum* L. extract had a median MIC of 22.086 (23.25, 20.33, 24.234, 20.53). 

## 4. Discussion

The antimicrobial activity of the tested *S. tectorum* L. extract ranged from 1.47 to 63.75 µg/mL, depending on the concentration of the extract and the tested microorganism. The tested concentrations showed a wide range of antimicrobial activity, which could be classified as high or moderate. *S. tectorum* L. extract was bacteriostatic at very high concentrations, or there was no bactericidal activity at most of the tested concentrations.

The results indicate that *S. tectorum* L. extract has a primarily inhibitory and bactericidal effect on microorganisms. The antimicrobial activity of dried leaf extracts was higher than that of fresh leaf extracts, indicating a higher concentration of antimicrobial compounds in the leaves. In addition, when antimicrobial activity against the tested bacterial species was observed, Gram-positive bacteria (*S. aureus*) were more susceptible. This is due to differences in the cell envelope composition of Gram-positive and Gram-negative bacteria, which affect the permeability and susceptibility of these organisms to different compounds [57,58,59].

Based on these results, we can conclude that ethanol extracted the compounds with the highest antimicrobial activity found in the plant’s leaves. This was expected, given that the leaves are described as the active plant parts in the ethnopharmacological usage of this plant. Furthermore, the obtained results can be attributed to phenolic and proanthocyanin compounds, which are well known for their antimicrobial activity [16,55,60,61]. Because of the slight discrepancy in results between antimicrobial activity and phenol composition (significant concentration in the roots but low antimicrobial activity), we can assume that other compounds, such as flavonoids (found in low concentrations in the leaf extract), play an important role in the antimicrobial effect of the tested extracts. In 2015, Rovanin et al. [62] used the Folin–Ciocalteu assay to determine TPC in extracts of *S. tectorum* (solvent extraction, solvent ethanol:water = 7:3) and other plants. The TPC of *S. tectorum* extract in the latter study was 16.00 mg GA/g extract. We found a much higher TPC of 139.42 mg GA/g when using ultrasound extraction. In the Rovanin et al. study, the leaves of *S. tectorum* L. were air-dried at 25–28 °C, which is most likely the reason for this difference. In our study, the leaves were lyophilized, allowing for a higher TPC to be preserved. In addition to TPC, the same authors reported PAC content, with the results expressed as mg of catechin equivalent (CE)/g extract [47]. They obtained a PAC value of 0.9 mg CE/g in an extract using a 7:3 ethanol:water mixture. Abram et al. [61] identified two major PAC compounds (4-dibenzyl-(-)-epigallocatechin and 4-dibenzyl-(-)-epigallocatechin-3-gallate) in *S. tectorum* extracts. Knez Marevci et al. [13] were the first to test and confirm the PAC content of dried *S. tectorum* extracts. They used conventional extraction methods and solvents to compare two lyophilized *S. tectorum* extracts, and obtained the following results using the same solvent mixture for all three methods: Soxhlet extraction yielded 13.40 mg PAC/g extract, cold solvent extraction yielded 15.89 mg PAC/g extract, and ultrasound extraction yielded 15.42 mg PAC/g extract. Several factors contribute to the increased interest in herbal medications in veterinary medicine. Among them, there is a widespread belief among the general public that medicinal plants are both practical and safer than synthetic compounds. When taken correctly, herbal medications often have few adverse effects, are well tolerated by most pets, and can be helpful in cases where conventional medications are not tolerated, or side effects are intolerable. Another primary reason is cost, as they are less expensive than traditional treatments [63,64,65]. Furthermore, they are regarded as a more sustainable approach. In this regard, phytotherapeutic remedies are also viable for treating otitis externa in dogs while avoiding synthetic drugs [66]. Plant-based medicine is becoming more popular in dogs due to its effectiveness and appropriate benefit–risk balance [32]. It may also aid in the treatment of subclinical or chronic diseases in the absence of conventional treatment. The widespread use of phytotherapy would also justify a discussion about the source of veterinarians’ knowledge in this field and increased academic training provided by veterinary faculties [66,67]. Otitis externa is one of the most common painful conditions in dogs. This a highly prevalent multifactorial skin disease that can be challenging to treat and accounts for up to 20% of pet counseling cases [68]. Bacterial reinfection becomes common if the root cause is not treated, needing prolonged antibiotic therapy [28,68]. The prolonged therapy increases the risk of antibiotic-resistant bacteria spreading from animals to humans and promotes their emergence. Resistance to antibiotics has emerged as one of the most serious public health concerns. This is how treatment failure appears to be coercing alternative approaches based on phytotherapeutic agents [32,69,70,71]. This study emphasizes the significance of antimicrobial resistance and the necessity for antibiotic alternatives, such as plant extract. As indicated in the results of our study, the *S. aureus* clinical isolate was resistant to the most frequently used or first-line antibiotic prescribed for otitis externa infection (gentamicin). Thus topical and oral antibiotics (gentamicin and enrofloxacin) are frequently used in canine bacterial ear infections, and resistance is a significant concern [72]. An increasing antimicrobial resistance in the *Staphylococcus* spp. and *P. aeruginosa*, isolated from dog ear clinical samples, has been reported [27,35]. 

Furthermore, this study opens the opportunity to initiate new investigations addressing some limitations of the current findings, especially regarding in vivo evaluation of the *Sempervivum tectorum* L. extract in the clinical cases of otitis externa in dogs and in vitro testing of other bacterial strains isolated from this disease.

## 5. Conclusions

Otitis externa in dogs is frequently complicated by the proliferation of numerous pathogenic bacterial strains. The treatment of this disorder in companion animals is primarily based on the use of synthetic antimicrobials. Increased resistance to potentially effective antimicrobials is among the adverse effects of these antimicrobials in veterinary medicine. Our findings indicate that *S. tectorum* L. has a high potential for use in pharmacy and phytotherapy. According to this information, parts of this plant are precious natural sources of antimicrobial substances.

Further research on this plant species should focus on a detailed qualitative analysis of its parts and in vivo studies of its medically active components to create a high-value natural pharmaceutical product.

## Figures and Tables

**Table 1 vetsci-10-00265-t001:** Total phenol and proanthocyanidin contents of *S. tectorum* L. extract.

Type of Extract	Proanthocyanidin Content (mg PAC/g Extract)	Total Phenol Content (mg GA/g Extract)
Ethanolic extract: 50% EtOH + 50% H_2_O (leaf)	15.39 ± 0.667	126.17 ± 0.334

All values are the average of three analyses ± SD.

**Table 2 vetsci-10-00265-t002:** Median minimal inhibitory concentration (MIC) and minimal bactericidal concentration (MBC) of most effective *S. tectorum* L. extract (µg/mL).

Bacterial Strain	Ethanolic Extract: 50% EtOH + 50% H_2_O (Leaf) (µg/mL)			Enrofloxacin/5 µg	Gentamycin/10 µg
	MIC	MBC	MBC/MIC Ratio	MIC	MIC
*S. aureus* clinical isolate	23.25	37.23	1.60	0.5	12.5
*S. aureus* ATCC 25923	20.33	37.29	1.83	0.25	1.125
*P. aeruginosa* clinical isolate	24.234	37.30	1.53	0.5	6.25
*P. aeruginosa* ATCC 27853	20.53	37.02	1.80	0.125	2.5

CLSI, 2020. *P. aeruginosa* breakpoints (µg/mL): enrofloxacin: ≤0.5 = susceptible, 1–2 = intermediate, ≥4 = resistant; gentamicin: ≤4 = susceptible, 4 = intermediate, ≥8 = resistant. *S. aureus* breakpoints (µg/mL): enrofloxacin: ≤0.5 = susceptible, 1–2 = intermediate, ≥4 = resistant; gentamicin: ≤4 = susceptible, 4 = intermediate, ≥8 = resistant.

**Table 3 vetsci-10-00265-t003:** Minimum inhibitory concentration of most effective *S. tectorum* extract against *S. aureus* and *P. aeruginosa* (clinical and standard) strains.

Plant Extract	Concentration of Extract (µg/mL)	*S. aureus* Clinical Isolate		*S. aureus* ATCC 25923		*P. aeruginosa* Clinical Isolate		*P. aeruginosa* ATCC 27853	
		MIC (µg/mL)	MBC (µg/mL)	MIC (µg/mL)	MBC (µg/mL)	MIC (µg/mL)	MBC (µg/mL)	MIC (µg/mL)	MBC (µg/mL)
*S. tectorum* L. ethanolic extract	0.5	0	0	0	0	0	0	0	0
	1.0	0	0	0	0	0	0	0	0
	2.0	0	0	1.47 ± 0.301	0	0	0	1.75 ± 0.25	0
	4.0	3.67 ± 0.381	0	3.53 ± 0.604	0	3.58 ± 0.378	0	3.66 ± 0.453	0
	8.0	7.5 ± 0.661	0	7.11 ± 0.808	0	7.7 ± 0.396	0	7.63 ± 0.436	0
	16.0	14.45 ± 0.5	15.93 ± 0.076	15.7 ± 0.396	15.98 ± 0.028	15.51 ± 0.475	15.96 ± 0.028	15.53 ± 0.604	15.75 ± 0.433
	32.0	30.05 ± 1.64	31.9 ± 0.132	30.83 ± 1.808	31.94 ± 0.06	30.95 ± 0.81	31.96 ± 0.02	30.86 ± 1.125	31.58 ± 0.381
	64.0	63.67 ± 0.381	63.86 ± 0.132	63.36 ± 0.583	63.95 ± 0.028	63.43 ± 0.419	64 ± 0.00	63.75 ± 0.25	63.73 ± 0.421

All values are the average of three analyses ± SD.

## Data Availability

Not applicable.

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
