# Peer review of "Antimicrobial Activity of *Sempervivum tectorum* L. Extract on Pathogenic Bacteria Isolated from Otitis Externa of Dogs"

_vetsci, 2023, doi:10.3390/vetsci10040265_

Round 1
Reviewer 1 Report (New Reviewer)
The manuscript is well written and easy to understand. The argument is of interest of the scientific community and fit the topics of Veterinary Sciences.
Some major flaws have been identified along the text. My major concern is related to the superficial argumentation about the clinical part.
L33against pathogenic bacteria isolated from ear swabs taken from dogs with otitis: it seems that bacteria were isolated from clinical cases of external otitis but in the abstract authors are referring to standard strains purchased by ATCC. Please, provide clear information in order not to mislead readers.
L64 … in Romania. Why it is important for Authors to specify the location? I know that the antimicrobial susceptibility can vary across countries or continents, thus if also Authors want to point out some peculiarities of the Romanian situation, they should explicit the concept here and also in the title.
L68-69: in human or veterinary medicine? In dogs/cats or whatever species? Please, specify.
L77-79: same specification as L68-69.
L86-87: also in this sentence, authors should provide more detailed information about the otitis externa. Are they referring to dogs? If yes, why is this pathology so relevant? Which are the remedies (allopathic or traditional) currently available? Risks/ benefits of both?
L88: not found but isolated.
L90: also, colonization by yeast (such as Malassezia spp.) could complicate the pathology and could represent an adjunctive challenge in therapeutic success. Please, improve this sentence.
L99: From western Romania: I would suggest removing this information from the aim of the study because authors will explain later and well the point. Please, specify here and in the abstract that you evaluate the antimicrobial activity “both in isolated bacteria and in standard strains”.
L201-202: specify the statistical methods that were applied, please.
L286: here authors mention another statistical software… and why authors are presenting in once again in paragraph 3.4 the statistical methods that it is supposed to be already explained in paragraph 2.5? Please, harmonize these two parts and include the results of statistical test in the other results removing paragraph 3.5.
Discussion: Authors completely missed the point to discuss the clinical implications of the use of phytotherapic compounds in otitis externa. They just provide some general comments (without references) from L327 till L346. Thus:
- Please provide the references
- L333-334: Authors are sure that plant extract cost less than traditional (allopathic) therapies? If yes, provide a reference, detailed information (may be this is a specific context related to Romania).
- What about AMR? Authors cited AMR in the introduction, but this topic is superficially treated along the discussion.
- - …increased academic training provided 342
by veterinary faculties… à also in this case, provide a reference and detail the information because this may be something peculiar of Romania (for example in my country, ethnopharmacology and phytotherapy are branches of veterinary medicine only for specialist like me…)
Hereby I suggest a list of papers that authors must consider improving the discussion:
- Domingo Ruiz-Cano, Ginés Sánchez-Carrasco, Amina El-Mihyaoui, Marino B. Arnao, Essential Oils and Melatonin as Functional Ingredients in Dogs, Animals, 10.3390/ani12162089, 12, 16, (2089), (2022).
- Tresch, M., Mevissen, M., Ayrle, H., Melzig, M., Roosje, P., & Walkenhorst, M. (2019). Medicinal plants as therapeutic options for topical treatment in canine dermatology? A systematic review. BMC Veterinary Research, 15, 174. https://doi.org/10.1186/s12917-019-1854-4
- Vercelli, C., Pasquetti, M., Giovannetti, G., Visioni, S., Re, G., Giorgi, M., Gambino, G., & Peano, A. (2021). In vitro and in vivo evaluation of a new phytotherapic blend to treat acute externa otitis in dogs. Journal of veterinary pharmacology and therapeutics, 44(6), 910–918. https://doi.org/10.1111/jvp.13000
- Sim, J., Khazandi, M., Pi, H., Venter, H., Trott, D. J., & Deo, P. (2019). Antimicrobial effects of cinnamon essential oil and cinnamaldehyde combined with EDTA against canine otitis externa pathogens. Journal of Applied Microbiology, 127(1), 99– 108.
As I wrote to Editors, I suggest major revision after a deep implementation of the discussion. In the present form, the manuscript is more a description of extraction method and antimicrobial susceptibility method, without emphasis on clinical outcomes.
Author Response
Dear Reviewer,
We thank you for your time, interest, and comments on our manuscript. We have thoroughly considered your comments and revised the text to reflect this. In a point-by-point fashion, we have addressed each comment below, trying to clarify each of the raised concerns. The new version of the manuscript contains all of the requested and operated changes highlighted using the Track Change option of Microsoft Word.
Thank you again for your time and consideration.
The manuscript is well written and easy to understand. The argument is of interest of the scientific community and fit the topics of Veterinary Sciences.
Some major flaws have been identified along the text. My major concern is related to the superficial argumentation about the clinical part.
Q1: L33 against pathogenic bacteria isolated from ear swabs taken from dogs with otitis: it seems that bacteria were isolated from clinical cases of external otitis but in the abstract authors are referring to standard strains purchased by ATCC. Please, provide clear information in order not to mislead readers.
Answer: I corrected in the revised version and mentioned the necessary details to clarify possible misunderstandings on the part of the readers (Lines 33-35).
Q2: L64 … in Romania. Why it is important for Authors to specify the location? I know that the antimicrobial susceptibility can vary across countries or continents, thus if also Authors want to point out some peculiarities of the Romanian situation, they should explicit the concept here and also in the title.
Answer: With respect to the reviewer's opinion, the composition and antimicrobial activity of Sempervivum depend on various factors such as the geographical origin, climatic conditions, method of collection, the quality of the processed plant raw material, treatment processes, and optimization of technological procedures for extracting the active components that have antimicrobial properties.
This aspect was also mentioned and can be found in the manuscript in Lines 243-244.
If we refer to the sensitivity of bacterial strains to antimicrobial substances, then it would be justified not to specify the location, as it is a matter of zonal, areal, or country-specificities.
We exemplify you with two publications that mention the place of origin of the plant:
Gentscheva G, Karadjova I, Minkova S, Nikolova K, Andonova V, Petkova N, Milkova-Tomova I. Optical Properties and Antioxidant Activity of Water-Ethanolic Extracts from Sempervivum tectorum L. from Bulgaria. Horticulturae. 2021; 7(12):520. https://doi.org/10.3390/horticulturae7120520.
Stojkovic, D.; Barros, L.; Petrovic, J.; Glamoclija, J.; Santos-Buelga, C.; Ferreira, I.C.F.R.; Sokovic, M. Ethnopharmacological uses of Sempervivum tectorum L. in southern Serbia: Scientific confirmation for the use against otitis linked bacteria. J. Ethnopharmacol. 2015, 176, 297–304. https://doi.org/10.1016/j.jep.2015.11.014.
Q3: L68-69: in human or veterinary medicine? In dogs/cats or whatever species? Please, specify.
Answer: As requested, we corrected in the revised version (Lines 69-71).
Q4: L77-79: same specification as L68-69.
Answer: As requested, we corrected in the revised version (Lines 81-89).
Q5: L86-87: also in this sentence, authors should provide more detailed information about the otitis externa. Are they referring to dogs? If yes, why is this pathology so relevant? Which are the remedies (allopathic or traditional) currently available? Risks/ benefits of both?
Answer: We thank you for the pertinent observations. In the revised form of the manuscript, we have inserted a paragraph to explain the relevant aspects of otitis externa in dogs and the motivation for the choice of the two bacterial species (Lines 104-111). Since it is a communication, we were limited in the number of pages, so we should have emphasized these details. Other information about otitis externa in dogs and the need to develop alternatives to conventional treatments are highlighted in Lines 372-386.
Q6: L88: not found but isolated.
Answer: As requested, we corrected in the revised version (Lines 69 and 71).
Q7: L90: also, colonization by yeast (such as Malassezia spp.) could complicate the pathology and could represent an adjunctive challenge in therapeutic success. Please, improve this sentence.
Answer: As requested, we corrected in the revised version (Lines 97-98).
Q8: L99: From western Romania: I would suggest removing this information from the aim of the study because authors will explain later and well the point. Please, specify here and in the abstract that you evaluate the antimicrobial activity “both in isolated bacteria and in standard strains”.
Answer: As requested, we corrected in the revised version (Line 113).
Q9: L201-202: specify the statistical methods that were applied, please.
Answer: As requested, we corrected in the revised version (Lines 218-221).
Q10: L286: here authors mention another statistical software… and why authors are presenting in once again in paragraph 3.4 the statistical methods that it is supposed to be already explained in paragraph 2.5? Please, harmonize these two parts and include the results of statistical test in the other results removing paragraph 3.5.
Answer: As requested, we corrected in the revised version (Lines 306-309): we removed paragraph 3.5 and added the information explained in paragraph 2.5 at the end discussion section.
Q11: Discussion: Authors completely missed the point to discuss the clinical implications of the use of phytotherapic compounds in otitis externa. They just provide some general comments (without references) from L327 till L346. Thus:
- Please provide the references
- L333-334: Authors are sure that plant extract cost less than traditional (allopathic) therapies? If yes, provide a reference, detailed information (may be this is a specific context related to Romania).
Answer: The use of medicinal plant extract is due to the saving of time in screening or chemical synthesis of random compounds, saving money, and dramatically reduced chances of toxicity – these being the most significant obstacles to the cost-effective discovery of drugs today. Instead, the results of this study drive the selection of medicinal plants for new drug discovery and provide information regarding the extraction of the active principle and the chemical substances from the plants identified as medicinal. This can lead to the discovery of lead compounds using modern research tools, which elucidate their structure, composition, and bioactivity. Herbal medicine might be an economical alternative treatment for otitis externa in dogs.
In the revised version of the manuscript, we have included three references to understand this aspect better:
Aminu Saleh Ahmad; Ruchi Sharma. "Comparitive Analysis of Herbal and Allopathic Treatment systems". European Journal of Molecular & Clinical Medicine, 7, 7, 2020, 2869-2876.
Karimi, A., Majlesi, M., & Rafieian-Kopaei, M. (2015). Herbal versus synthetic drugs; beliefs and facts. Journal of nephropharmacology, 4(1), 27–30.
Romero, B., Susperregui, J., Sahagún, A. M., Diez, M. J., Fernández, N., García, J. J., López, C., Sierra, M., & Díez, R. (2022). Use of medicinal plants by veterinary practitioners in Spain: A cross-sectional survey. Frontiers in veterinary science, 9, 1060738. https://doi.org/10.3389/fvets.2022.1060738.
- What about AMR? Authors cited AMR in the introduction, but this topic is superficially treated along the discussion.
Answer: In the revised manuscript, we discussed the problem of antibiotic resistance in the context of otitis externa in dogs (Lines 375-386).
- - …increased academic training provided 342
by veterinary faculties… à also in this case, provide a reference and detail the information because this may be something peculiar of Romania (for example in my country, ethnopharmacology and phytotherapy are branches of veterinary medicine only for specialist like me…)
Answer: It is not a specific situation for veterinary medical education in Romania; there are worldwide concerns about creating a curriculum at the level of specialized faculties for integrative veterinary medical education. Within our faculty is a Department of Ethnobotany and Toxicology, which is based on the knowledge of medicinal plants from Romania, with the active principles and the problem of toxic effects and very little information about phytotherapy. There is only one optional discipline in one semester related to the importance of phytotherapy and its use in pets and farm animals.
Thus, in the revised version of the manuscript, we have included two bibliographic sources in this paragraph.
Memon, M. A., Shmalberg, J., Adair, H. S., 3rd, Allweiler, S., Bryan, J. N., Cantwell, S., Carr, E., Chrisman, C., Egger, C. M., Greene, S., Haussler, K. K., Hershey, B., Holyoak, G. R., Johnson, M., Jeune, S. L., Looney, A., McConnico, R. S., Medina, C., Morton, A. J., Munsterman, A., Wakshlag, J. (2016). Integrative veterinary medical education and consensus guidelines for an integrative veterinary medicine curriculum within veterinary colleges. Open veterinary journal, 6(1), 44–56. https://doi.org/10.4314/ovj.v6i1.7.
Memon, M. A., Shmalberg, J. W., & Xie, H. (2021). Survey of Integrative Veterinary Medicine Training in AVMA-Accredited Veterinary Colleges. Journal of veterinary medical education, 48(3), 289–294. https://doi.org/10.3138/jvme.2019-0067.
Hereby I suggest a list of papers that authors must consider improving the discussion:
Answer: Thank you for your pertinent comments. We considered your recommendations and included in the revised version a paragraph in the discussion chapter about clinical aspects, resistance to antibiotics, and the use of medicinal plants in the therapeutic management of external otitis in dogs (Lines 361-364; 372-386).
We have also included the suggested and other relevant references:
Domingo Ruiz-Cano, Ginés Sánchez-Carrasco, Amina El-Mihyaoui, Marino B. Arnao, Essential Oils and Melatonin as Functional Ingredients in Dogs, Animals, 10.3390/ani12162089, 12, 16, (2089), (2022).
Tresch, M., Mevissen, M., Ayrle, H., Melzig, M., Roosje, P., & Walkenhorst, M. (2019). Medicinal plants as therapeutic options for topical treatment in canine dermatology? A systematic review. BMC Veterinary Research, 15, 174. https://doi.org/10.1186/s12917-019-1854-4.
Vercelli, C., Pasquetti, M., Giovannetti, G., Visioni, S., Re, G., Giorgi, M., Gambino, G., & Peano, A. (2021). In vitro and in vivo evaluation of a new phytotherapic blend to treat acute externa otitis in dogs. Journal of veterinary pharmacology and therapeutics, 44(6), 910–918. https://doi.org/10.1111/jvp.13000.
Sim, J., Khazandi, M., Pi, H., Venter, H., Trott, D. J., & Deo, P. (2019). Antimicrobial effects of cinnamon essential oil and cinnamaldehyde combined with EDTA against canine otitis externa pathogens. Journal of Applied Microbiology, 127(1), 99– 108.
Bourély, C., Cazeau, G., Jarrige, N., Leblond, A., Madec, J. Y., Haenni, M., & Gay, E. (2019). Antimicrobial resistance patterns of bacteria isolated from dogs with otitis. Epidemiology and infection, 147, e121. https://doi.org/10.1017/S0950268818003278.
Kwon J, Yang M-H, Ko H-J, Kim S-G, Park C, Park S-C. Antimicrobial Resistance and Virulence Factors of Proteus mirabilis Isolated from Dog with Chronic Otitis Externa. Pathogens. 2022; 11(10):1215. https://doi.org/10.3390/pathogens11101215.
Bertelloni, F., Cagnoli, G., & Ebani, V. V. (2021). Virulence and Antimicrobial Resistance in Canine Staphylococcus spp. Isolates. Microorganisms, 9(3), 515. https://doi.org/10.3390/microorganisms9030515.
Park S, Oh T, Bae S. The stability and in vitro antibacterial efficacy of enrofloxacin and gentamicin solutions against Staphylococcus pseudintermedius over 28 days. Vet Dermatol. 2023 Feb;34(1):28-32. doi: 10.1111/vde.13131.
As I wrote to Editors, I suggest major revision after a deep implementation of the discussion. In the present form, the manuscript is more a description of extraction method and antimicrobial susceptibility method, without emphasis on clinical outcomes.

Reviewer 2 Report (New Reviewer)
The paper reports the composition and the antibcterial activity of ethanolic extracts of Sempervivum tectorum, against bacterial isolates from canine external otitis. The manuscript is well written and can be considered for publication, however, the section regarding microbiological aspects should be implemented. The Authors don't provide any information about the choice of target bacteria, nor of the impact of canine external otitis in veterinary clinical practice. The only mention I have found is in conclusion section, so I suggest the Authors to enhance introduction and discussion sections to point out the clinical problem (this is a contribute for a special issue of Veterinary Sciences) and to clarify the need of such potential remedia in the treatment. Data concerning the setting up of microdilution testing should be provided, too (i.e.have the Authors followed CLSI guidelines?).
Author Response
Dear Reviewer,
Thanks for taking the time to review our manuscript and for your close attention to detail. We highly appreciate your suggestions and comments, which significantly improved the quality of the submission. During the revision process, we tried to do our best to address each of these successfully. Please see below our responses in a point-by-point fashion to the raised concerns.
The paper reports the composition and the antibcterial activity of ethanolic extracts of Sempervivum tectorum, against bacterial isolates from canine external otitis. The manuscript is well written and can be considered for publication, however, the section regarding microbiological aspects should be implemented.
Q1: The Authors don't provide any information about the choice of target bacteria, nor of the impact of canine external otitis in veterinary clinical practice. The only mention I have found is in conclusion section, so I suggest the Authors to enhance the introduction and discussion sections to point out the clinical problem (this is a contribute for a special issue of Veterinary Sciences) and to clarify the need of such potential remedia in the treatment.
Answer: Thanks for the comments and suggestions. We have developed this aspect in the revised manuscript version (Lines 372-386).
Q2: Data concerning the setting up of microdilution testing should be provided, too (i.e.have the Authors followed CLSI guidelines?).
Answer: Concerning the reviewer's opinion, this concern was already described in the original submitted version, between lines 178-179 (CLSI M07-A9; CLSI. Methods for Dilution Antimicrobial Susceptibility Tests for Bacteria That Grow Aerobically; Approved Standard—11th Edition. CLSI document M07-A9. Wayne, PA: Clinical and Laboratory Standards Institute; 2018).

Round 2
Reviewer 1 Report (New Reviewer)
Dear Authors, The manuscript has been significantly improved.
I have only two minor comments:
- L4-96: provide here the contracted forms of the bacteria names. Thus, you can use contracted forms along the manuscript.
-paragraph 3.3: something went wrong with layout. Try to fix it but I'm sure that Editorial office will help you to fix it along the proof-editing process.
Author Response
Dear Reviewer,
Thanks for taking the time to review our manuscript and for your close attention to detail. We highly appreciate your suggestions and comments, which significantly improved the quality of the submission.
Dear Authors, The manuscript has been significantly improved.
I have only two minor comments:
Q1:L4-96: provide here the contracted forms of the bacteria names. Thus, you can use contracted forms along the manuscript.
Answer: Thank you for the relevant recommendations, which lead to the improvement of the quality of the manuscript. We standardized the names of the bacterial species in the revised manuscript, so we used the full name when it was first used and contracted forms along in the manuscript. As requested, we corrected in the revised version (Lines 34-35; 44-45; 95; 108; 178-179).
Q2: paragraph 3.3: something went wrong with layout. Try to fix it but I'm sure that Editorial office will help you to fix it along the proof-editing process.
Answer: As requested, we corrected in the revised version (Lines 272-282).

Reviewer 2 Report (New Reviewer)
The Authors have followed all the suggestions and the manuscript can be now accepted
Author Response
We want to thank Reviewer for taking the time and effort to review the manuscript. We sincerely appreciate all your valuable comments and suggestions, which helped us improve the manuscript's quality.
This manuscript is a resubmission of an earlier submission. The following is a list of the peer review reports and author responses from that submission.
Round 1
Reviewer 1 Report
The work is described very well in the materials and methods. The plant is known for its pain-relieving and astringent properties, but the evidence of antibacterial activity is not so evident and above all tested in vitro and only on 2 laboratory strains and on only 2 clinical bacterial isolates (too few).
Author Response
Answer for reviewer 1
Q1: The work is described very well in the materials and methods. The plant is known for its pain-relieving and astringent properties, but the evidence of antibacterial activity is not so evident and above all tested in vitro and only on 2 laboratory strains and on only 2 clinical bacterial isolates (too few).
Answer: Chemical composition and biological activities (antinociceptive, liver-protecting, membrane stabilizing effect, antimicrobial, anti-inflammatory, and antioxidant capacity) of a genus of Sempervivum L. have been documented and reported in the literature. Primarily the investigations of the antimicrobial activity of extracts and juice of Sempervivum tectorum L. were performed in recent studies.
- Kazlagić, A., Lagumdžija, A., Borovac, B., Hamidović, S., Abud, O.A., Omanović-Mikličanin, E. (2020). Green Synthesis and Characterization of Silver Nanoparticles Using Fresh Leaf Extract of Aloe vera barbadensis Miller, Aloe veraand Sempervivum tectorumand Its Antimicrobial Activity Studies. In: Brka, M., Omanović-Mikličanin, E., Karić, L., Falan, V., Toroman, A. (eds) 30th Scientific-Experts Conference of Agriculture and Food Industry. AgriConf 2019. IFMBE Proceedings, vol 78. Springer, Cham. https://doi.org/10.1007/978-3-030-40049-1_42.
- Karabegović, I. T., Stojičević, S. S., Veličković, D. T., Nikolić, N. Č., & Lazić, M. L. (2018). Direct ultrasound-assisted extraction and characterization of phenolic compounds from fresh houseleek (Sempervivum marmoreum L.) leaves. HEMIJSKA INDUSTRIJA (Chemical Industry), 72(1), 13–21. https://doi.org/10.2298/HEMIND170402017K.
- Stojković, D., Barros, L., Petrović, J., Glamoclija, J., Santos-Buelga, C., Ferreira, I. C., & Soković, M. (2015). Ethnopharmacological uses of Sempervivum tectorum L. in southern Serbia: Scientific confirmation for the use against otitis linked bacteria. Journal of ethnopharmacology, 176, 297–304. https://doi.org/10.1016/j.jep.2015.11.014.
Ethnopharmacological use of S. tectorum juice for treating ear pain is justified since the juice possesses antimicrobial activity towards clinical isolates of bacteria linked to otitis. Antimicrobial activity was tested on bacteria isolated from ear swabs of the patients suffering from ear pain (otitis).
- Rovcanin, B.R.; Cebovic, T.; Steševic, D.; Kekic, D.; Ristic, M. Antibacterial effect of Herniaria hirsuta, Prunus avium, Rubia tinctorum and Sempervivum tectorum plant extracts on multiple antibiotic resistant Escherichia coli. Biosci. J. 2015, 31, 1852–1861.
In the present work, Rovcanin et col., was examined the antibacterial effect of following plants (Herniaria hirsuta, Prunus avium, Rubia tinctorum and Sempervivum tectorum). The bacterial model used for estimation of bacterial susceptibility is hospital multiple antibiotic resistant E. coli strain. E. coli ATCC 25922 was used for standard comparison of bacterial susceptibility. Concentration of total phenols, flavonoids, tannins, antocyanins and saponins was determined in plant extracts. Aqueous extracts of R. tinctorum and S. tectorum have higher antibacterial potential than theirs ethanol extracts.
- Abram, V., & Donko, M. (1999). Tentative identification of polyphenols in Sempervivum tectorum and assessment of the antimicrobial activity of Sempervivum L.Journal of agricultural and food chemistry, 47(2), 485–489. https://doi.org/10.1021/jf980669d.
Polyphenols were isolated from sliced fresh leaves of Sempervivum tectorum. Antimicrobial activity of Sempervivum L. leaves against six of seven selected microorganisms was observed.
Based on the methodology described by other authors, in the case of testing some plant extracts, the purpose of the research was not to establish the number of sensitive strains but to determine the MIC of the plant extract (DOI: 10.1016/j.sjbs.2017.02.004, https://doi.org/10.5897/JMPR12.239, DOI: 10.31383/ga.vol6iss1pp38-51, https://doi.org/10.31083/j.fbe1404025). In other studies, they used only standardized strains of different bacterial species. And in our research, we followed these results.
Furthermore, this study opens the opportunity to initiate new investigations addressing some limitations of the current findings, especially regarding in vivo evaluation of the Sempervivum tectorum L. extract in the clinical cases of otitis externa in dogs and in vitro testing of other bacterial strains isolated from these diseases.
With due respect for your hard work and expertise,
Lecturer DVM Ph.D., MSc DEGI Janos

Reviewer 2 Report
Authors have done study on antimicrobial activity of Sempervivum tectorum L. extract on pathogenic bacteria isolated from otitis externa of dog.
Following queries need to be address first:
1) Authors did not mention howmany clinical isolates were used ? please provide list and individual data and its result in supplymentary files.
2) Authors are requested to add photograph/raw data/any realted materials related to your experiment in supplymentary files.
3) Abstract part should include antimicrobial result by adding two-three lines.
4) Discussion part need more comparative analysis and add more informative conclusion regarding further usage and how it can be applicable to the veterinary clinical use.
Author Response
Answer for reviewer 2
Authors have done study on antimicrobial activity of Sempervivum tectorum L. extract on pathogenic bacteria isolated from otitis externa of dog.
Following queries need to be address first:
Q1: Authors did not mention how many clinical isolates were used? please provide list and individual data and its result in supplymentary files.
Answer: Our study used two clinical strains (Staphylococcus aureus and Pseudomonas aeruginosa) isolated from otitis externa in dogs. I did not find data on the number of strains needed to perform the tests in the specialized literature (DOI: 10.1016/j.sjbs.2017.02.004, https://doi.org/10.5897/JMPR12.239, DOI: 10.31383/ga.vol6iss1pp38-51, https://doi.org/10.31083/j.fbe1404025). Based on the methodology described by other authors, in the case of testing some plant extracts, the purpose of the research was not to establish the number of sensitive strains but to determine the MIC and MBC of the plant extract. In other studies, they used only standardized strains of different bacterial species. And in our study, we followed these results. Press release research is underway to establish the in vivo action, respectively, the in vitro testing on several bacterial isolates. We also attach a document (Supplementary file 1) containing the primary data, which we also included in the revised manuscript (line 322).
Q2: Authors are requested to add photograph/raw data/any realted materials related to your experiment in supplymentary files.
Answer: We highly appreciate your suggestions and comments, which significantly improved the quality of the submission. I have attached a document with photos that contain details regarding the process of testing the extract of Sempervivum tectorum L. (Supplementary file 2), which we also included in the revised manuscript (line 139).
Q3: Abstract part should include antimicrobial result by adding two-three lines.
Answer: We formulated a few paragraphs, included in the revised version of the manuscript (Lines 49-55).
ʺThe antimicrobial activity of the tested S. tectorum L. extracts ranged from 1.47 to 63.75 µg/ml, starting with 1.47 µg/ml and 1.75 µg/ml against S. aureus ATCC 25923 and P. aeruginosa ATCC 27853 strains, respectively. Likewise, S. tectorum L. ethanol extract demonstrated a bacteriostatic effect against S. aureus clinical isolate with a median MIC of 23.25 µg/ml and MBC of 37.23 µg/ml; and bactericidal against S. aureus ATCC 25923 with the median MIC of 20.33 µg/ml and MBC of 37.29 µg/ml. In the Gram-negative P. aeruginosa clinical and standard strains, the expressed MIC and MBC values were 24.234 and 20.53 µg/ml for MIC, and 37.30 and 37.02 µg/ml for MBC, respectively. ʺ
Q4: Discussion part need more comparative analysis and add more informative conclusion regarding further usage and how it can be applicable to the veterinary clinical use.
Answer: Thank you for your valuable time in revising this manuscript. We appreciate the pertinent observations, to which we have responded punctually to facilitate the understanding of the unclear aspects. We formulated a few paragraphs, included in the revised version of the manuscript (Lines 395-406).
ʺSeveral factors contribute to the increased interest in herbal medications in veterinary medicine. Among them, there is a widespread belief among the general public that medicinal plants are both practical and safer than synthetic compounds. Another primary reason is cost, as they are less expensive than traditional treatments. Furthermore, they are regarded as a more sustainable approach. In this regard, phytotherapeutic remedies are also viable for treating otitis externa in dogs while avoiding synthetic drugs.
Plant-based medicine is becoming more popular in dogs due to its effectiveness and appropriate benefit-risk balance. It may also aid in the treatment of subclinical or chronic diseases in the absence of conventional treatment. The widespread use of phytotherapy would also justify a discussion about the source of veterinarians' knowledge in this field and increased academic training provided by veterinary faculties.
Furthermore, this study opens the opportunity to initiate new investigations addressing some limitations of the current findings, especially regarding in vivo evaluation of the Sempervivum tectorum L. extract in the clinical cases of otitis externa in dogs and in vitro testing of other bacterial strains isolated from this diseasesʺ.
To be easily findable in the revised manuscript we marked all our answers/corrections in blue.
With due respect for your hard work and expertise,
Lecturer DVM Ph.D., MSc DEGI Janos

Reviewer 3 Report
Line 192: "first, 800 L of sodium carbonate 192 (7.5% w/v) was added to 20 mL of each sampl..." Obviously a mistake, should be 800 mL?
Lines 206-207: "The extracts were tested against a panel of microorganisms, including Gram-positive 206 Staphylococcus aureus ATCC 25923 and clinical isolate, and Gram-negative Pseudomonas 207 aeruginosa ATCC 27853" An explanation why authors decided to test S.aureus should be added. S.aureus is a human pathogen and rarely is involved in OE of dogs while S.pseudintermedius is commonly involved in OE of dogs. Testing against the latter would be most interesting. Also concerning Lines 336-340.
Line 214: nrofloxacin should be corrected to norfloxacin
Lines 337-338: "P. aeruginosa is one of the most common pathogenic bacteria responsible for otitis externa in dogs".It is suggested that instead of "responsible" word "involved" is used, since bacteria are not primary cause of OE. They are secondary cause.
Lines 354-355: Sentence "P. aeruginosa is one of the 337 most common pathogenic bacteria responsible for otitis externa in dogs." is not clear, please explain.
Lines 391-392: "Otitis externa in dogs is frequently caused by the proliferation of numerous pathogenic bacterial strains" instead of "caused" word "complicated" is suggested to be used.
Discussion would be interesting on the obvious differences in MIC between clinical strains of bacteria compared to standard strains (Table 2). It would be interesting to know wether tested antibiotics were used in the treatment of dogs from whom material was taken for testing.
Author Response
Answer for reviewer 3
Q1: Line 192: "first, 800 L of sodium carbonate 192 (7.5% w/v) was added to 20 mL of each sampl..." Obviously a mistake, should be 800 mL?
Answer: Thank you for reporting this error. We corrected it in the revised manuscript.
Q2: Lines 206-207: "The extracts were tested against a panel of microorganisms, including Gram-positive 206 Staphylococcus aureus ATCC 25923 and clinical isolate, and Gram-negative Pseudomonas 207 aeruginosa ATCC 27853" An explanation why authors decided to test S.aureus should be added. S.aureus is a human pathogen and rarely is involved in OE of dogs while S.pseudintermedius is commonly involved in OE of dogs. Testing against the latter would be most interesting. Also concerning Lines 336-340.
Answer: Indeed, the involvement of Staphylococcus aureus bacteria in the pathogenesis of otitis externa in dogs is less important, compared to the representative species for dogs, Staphylococcus pseudintermedius (isolation percentage from clinical cases 4.9-10.8% vs. 35-65%). Staphylococcus pseudintermedius and Staphylococcus aureus are common colonizers of companion animals, but they are also considered opportunistic pathogens, causing diseases of diverse severity. Canine S. aureus isolates tend to be more resistant than human isolates, with significant differences in the frequency of resistance to several antibiotics, with public health importance. This aspect possibly reflecting higher use of antibiotics in veterinary practice. Regarding Staphylococcus aureus, molecular characterization has revealed that companion animals are colonized or infected by hospital-associated and community-associated MRSA clones from humans in close contact, which suggests an anthropozoonotic origin. The presence to MRSA in the clinical setting, has become a worldwide problem in animal health. It is frequently associated with a multidrug resistance phenotype, which limits the therapeutic options for veterinarians.
Furthermore, this study opens the opportunity to initiate new investigations addressing some limitations of the current findings, especially regarding in vivo evaluation of the Sempervivum tectorum L. extract in the clinical cases of otitis externa in dogs and in vitro testing of other bacterial strains isolated from these diseases.
Q3: Line 214: nrofloxacin should be corrected to norfloxacin
Answer: Thank you for the pertinent observation, it is about enrofloxacin. We have corrected the error in the revised manuscript.
Q4: Lines 337-338: "P. aeruginosa is one of the most common pathogenic bacteria responsible for otitis externa in dogs".It is suggested that instead of "responsible" word "involved" is used, since bacteria are not primary cause of OE. They are secondary cause.
Answer: We appreciate the helpful recommendation. We replaced the word responsible with involved in the revised manuscript.
Q5: Lines 354-355: Sentence "P. aeruginosa is one of the 337 most common pathogenic bacteria responsible for otitis externa in dogs." is not clear, please explain.
Answer: The presence of the number 337 is a drafting error. It has no relevance to the testing of Pseudomonas aeruginosa strains. We have corrected the error in the revised manuscript.
Q6? Lines 391-392: "Otitis externa in dogs is frequently caused by the proliferation of numerous pathogenic bacterial strains" instead of "caused" word "complicated" is suggested to be used.
Answer: We appreciate the helpful recommendation. We replaced the word caused with complicated in the revised manuscript.
Q7: Discussion would be interesting on the obvious differences in MIC between clinical strains of bacteria compared to standard strains (Table 2). It would be interesting to know wether tested antibiotics were used in the treatment of dogs from whom material was taken for testing.
Answer: In recent years, several reports have evidenced an increase in the resistance rates for some important antimicrobials, such as fluoroquinolones, in Staphylococcus spp. isolates recovered from companion animals in European countries.
The two tested antibiotics were chosen because they are found in commercial topical products, used in the treatment of otitis externa in dogs. The availability of these commercial topical products is extremely high, animal owners can obtain these solutions without the need for a medical prescription and not always on the recommendation of the clinician. Recently a study showed that the two antibiotics in combination with an anti-inflammatory has a considerable effect.
Solutions of 1% enrofloxacin and 0.3% gentamicin in normal saline with 0.1% dexamethasone maintained chemical stability and bactericidal efficacy over 28 days. These solutions can be considered as alternatives to commercial preparations for treatment of canine OE when indicated (https://doi.org/10.1111/vde.13131).
With due respect for your hard work and expertise,
Lecturer DVM Ph.D., MSc DEGI Janos

Round 2
Reviewer 1 Report
-
Reviewer 2 Report
Thank you for addressing all questions and adding supportive documents in supplementary files.